# Brisk Walking Pace Is Associated with Better Cardiometabolic Health in Adults: Findings from the Chilean National Health Survey 2016–2017

**DOI:** 10.3390/ijerph20085490

**Published:** 2023-04-12

**Authors:** Igor Cigarroa, Michelle Bravo-Leal, Fanny Petermann-Rocha, Solange Parra-Soto, Yeny Concha-Cisternas, Carlos Matus-Castillo, Jaime Vásquez-Gómez, Rafael Zapata-Lamana, María Antonia Parra-Rizo, Cristian Álvarez, Carlos Celis-Morales

**Affiliations:** 1Escuela de Kinesiología, Facultad de Salud, Universidad Santo Tomás, Los Ángeles 4440000, Chile; 2Centro de Investigación Biomédica, Facultad de Medicina, Universidad Diego Portales, Santiago 8370068, Chile; 3Department of Nutrition and Public Health, Universidad del Bío-Bío, Chillan 3780000, Chile; 4School Cardiovascular and Metabolic Health, University of Glasgow, Glasgow G12 8TA, UK; 5Pedagogía en Educación Física, Facultad de Educación, Universidad Autónoma de Chile, Talca 3460000, Chile; 6Departamento de Ciencias del Deporte y Acondicionamiento Físico, Universidad Católica de la Santísima Concepción, Concepción 4030000, Chile; 7Centro de Investigación de Estudios Avanzados del Maule (CIEAM), Universidad Católica del Maule, Talca 3460000, Chile; 8Laboratorio de Rendimiento Humano, Grupo de Estudios en Educación, Actividad Física y Salud (GEEAFyS), Universidad Católica del Maule, Talca 3460000, Chile; 9Escuela de Educación, Universidad de Concepción, Los Ángeles 4440000, Chile; 10Faculty of Health Sciences, Valencian International University—VIU, 46002 Valencia, Spain; 11Department of Health Psychology, Faculty of Social and Health Sciences, Campus of Elche, Miguel Hernandez University (UMH), 03202 Elche, Spain; 12Exercise and Rehabilitation Sciences Institute, School of Physical Therapy, Faculty of Rehabilitation Sciences, Universidad Andres Bello, Santiago 7591538, Chile

**Keywords:** walking pace, glycaemia, glycosylated hemoglobin A, blood pressure, health surveys, Chile

## Abstract

Background: Although the importance of walking for promoting a better cardiometabolic health is widely known (this includes both cardiovascular and metabolic/endocrine systems), there is little knowledge regarding its appropriate pace to provide adults with more cardiometabolic benefits. Aim: To analyze the associations between different walking pace categories and cardiometabolic health markers in the adult Chilean population. Methods: Cross-sectional study. A total of 5520 participants aged 15 to 90 years old from the Chilean National Health Survey (CNHS) 2016–2017 were included. Walking pace categories (slow, average, and brisk) were collected through self-reported methods. Glycaemia, glycosylated hemoglobin (HbA1c), gamma glutamyl transferase (GGT), vitamin D2, vitamin D3, systolic and diastolic blood pressure, and lipid profile (Total, HDL, LDL, VLDL, No HDL cholesterol and triglycerides) were determined using blood sample tests and measured with the standardized methods described in the CNHS 2016–2017. Results: People who had a brisk walking pace were associated with lower levels of glycaemia, HbA1c, GGT, systolic and diastolic blood pressure, and higher vitamin D3 levels compared with those with a slow walking pace. Moreover, people with a brisk walking pace had lower levels of VLDL cholesterol compared with those with a slow walking pace. However, after adjusting the model to include sociodemographic background, nutritional status, and lifestyle variables, the differences remained only for glycaemia, HbA1c and systolic blood pressure levels. Conclusions: A brisk walking pace was associated with better cardiometabolic health markers and lipid profile compared with a slow walking pace.

## 1. Introduction

Lifestyle changes and demographic and socioeconomic changes globally have been one of the causes of changes in epidemiological profiles of the population worldwide. They have also caused an increase in cardiovascular diseases [1]. Diabetes, dyslipidemia, and arterial hypertension are the main chronic non-communicable diseases that can lead to death, accounting for 70% of deaths globally and challenging health systems to implement, strengthen and redesign health policies to address this [2]. Arterial hypertension, in addition to altered serum glucose, lipid concentrations (i.e., total cholesterol [TC], low-density lipoprotein [LDL-C], high-density lipoprotein [HDL-C], and triglycerides [TG]), alcohol use, smoking, a sedentary lifestyle, physical inactivity [3], and obesity [4] are the main cardiovascular risk factors that increase non-communicable diseases. In the Americas, cardiovascular diseases are responsible for 4 out of 5 annual deaths in people aged from 30 to 69 years old, and an increase in mortality is expected in the coming decades due to population growth, aging, land developments, exposure to the environment, and other risk factors. Thirty-five percent of deaths caused by cardiovascular and metabolic disorders occur in the population aged between 30 and 70 [5]. In the case of Chile, 33.3% of the population have smoking habits, 86.7% are physically inactive, 39.8% are overweight, 31.2% are obese, 27.6% have arterial hypertension, and 12.3% report having type 2 diabetes [6]. 

In populations with different ages [7] and underlying health conditions, a variety of health markers—such as aerobic capacity and muscle strength [8]—and a variety of assessment tests—such as grip strength, timed up-and-go, standing balance, knee-extension strength, and gait speed [9,10]—are used to measure and report physical health. Among these markers of physical health, walking pace has been recognized as a well-established predictive biomarker of life expectancy, risk of disability, health outcomes, and mortality [11,12,13,14]. Walking pace is an indicator capable of predicting the state of health and the risk of functional decline, as well as other health indicators such as days of hospitalization, the risk of disability, the level of care needed when discharged from hospital, and mortality [11,15]. Walking pace assessed at a normal pace [>1 m/second (m/s)] [16] has been shown to be a reliable, sensitive, valid, and specific measure to report health condition in adults [15]. 

The scientific evidence establishes walking pace as a predictor of health status [17,18]. However, to date, there have been few studies in Latin American countries using walking pace as a marker of cardiometabolic health [19,20]. This study analyzed the data from the Chilean National Health Survey (CHNS) 2016–2017, which carried out household surveys in a sample of the Chilean population. The results obtained in this research provide essential information when considering walking pace as a tool for assessing cardiometabolic risk in the general population. The aim of this study was to analyze the associations between different walking pace categories and markers of cardiometabolic health in the adult Chilean population.

## 2. Materials and Methods

### 2.1. Study Design

Non-experimental study, analytical and cross-sectional design. This is a secondary study from the CNHS 2016–2017. The CNHS 2016–2017 was conducted within Chilean households with participants aged from 15 to 90 years old. From the original simple size (*n* = 6233) who took part in the CNHS 2016-2017, only (*n* = 5520) participants were available in the data to study outcomes and covariates, and they were included in all the analyses. The CNHS 2016–2017 was funded by the Chilean Ministry of Health and approved by the Ethics Research Committee of School of Medicine at the Pontificia Universidad Católica de Chile (code n°16-019). All participants provided written consent before participation [21]. Additionally, the study was conducted in accordance with the Declaration of Helsinki.

### 2.2. Variables and Measuring Instruments

#### 2.2.1. Walking Pace

Self-reported walking pace was determined with the question “How would you describe your usual walking pace?” Participants were asked to select one of the following walking pace categories to answer the question: slow, average, or brisk pace. 

#### 2.2.2. Cardiometabolic Health Markers and Lipid Profile

The participants’ blood samples were obtained by a trained nurse after a fasting period, following nationally standardized protocols [20]. Cardiometabolic health markers (glycaemia, glycosylated hemoglobin (HbA1c), gamma glutamyl transferase (GGT), vitamin D2, vitamin D3, systolic blood pressure, diastolic blood pressure, and lipid profile [TC, HDL-C, LDL-C, very low-density lipoprotein (VLDL-C), No HDL-C and TG]) were measured with standardized methods previously described in the CNHS 2016–2017 [21]. 

Blood glucose levels were determined with HbA1c (glycosylated hemoglobin A) and glycaemia levels from a blood sample, which was drawn by trained nurses after 11 h of fasting by participants. The results were classified according to the parameters proposed by the American Diabetes Association (ADA)—HbA1c: 5.7%: normal, HbA1c: 5.7–6.4%: pre-diabetes, HbA1c greater than 6.5%: T2D; and glycaemia: >100 to 125 mg/dL: pre-diabetes, glycemia: >126 mg/dL: diabetes) [22].

#### 2.2.3. Sociodemographic, Lifestyle, and General Health Variables

##### Sociodemographic Variables

Age, sex (male or female), age group (15–37, 37–56, or >56 years old), educational level (primary <8, secondary 8 to 12, or higher education >12 years), area of residence (rural or urban) were determined using the questionnaires of the CHNS 2016–2017 [21].

##### Lifestyle Variables

Smoking habits, alcohol use (Alcohol Use Disorders Identification Test (AUDIT)), salt, fruit and vegetable intake, hours of sleep, self-perception of health, and personal wellbeing were also obtained using the questionnaires of the CHNS 2016–2017.

##### Physical Activity (PA)

The time allocated for PA related to commuting activities (walking, cycling) and for PA with moderate or vigorous intensity during leisure and work were obtained using the GPAQ analysis guide (Global Physical Activity Questionnaire v2) [21,22]. To estimate total PA, the variables were expressed in METs (Metabolic equivalent of Task). An energy expenditure < 600 MET/minutes/week— or its equivalent of 150 min of moderate to vigorous intensity of PA, or 75 min of vigorous intensity of PA per week, or a combination of the two—is considered to be a cut-off point for physical inactivity, according to the World Health Organization (WHO)’s recommendations and the GPAQ analysis guide’s specifications [23]. 

Additionally, sedentary behavior was determined using the time (hours) allocated to activities that involve sitting or reclining during free time or work (sitting time at the computer, watching TV, travelling by bus, train, car, etc.) [23,24]. 

##### Anthropometric Measures 

Weight was measured using a digital scale (TANITA Model HD-313^®^) and height was measured with a height rod in participants’ homes. Participants did not wear shoes and were dressed in light clothing for both measurements. Weight and height measurements were both carried out by trained nurses or midwives using standardized methods. Nutritional status and body fat were determined according to body mass index (BMI) based on the WHO’s cut-off points: underweight: <18.5 kg/m^2^; normal weight: 18.5–24.9 kg/m^2^; overweight: 25.0–29.9 kg/m^2^; and obesity: ≥30.0 kg/m^2^. In addition, central obesity was defined as waist circumference (WC) ≥ 88 cm for women and ≥102 cm for men [25].

### 2.3. Statistical Analyses

The analyses were performed by weighting the survey to the total national population, as suggested in the CNHS 2016–2017. The characterization data were presented as weighted means for continuous variables and as a weighted prevalence for categorical variables with their corresponding 95% confidence intervals (95% CI). Then, the associations of walking pace categories with cardiometabolic health and lipid profile markers were investigated using linear regression analyses. Data were presented as mean scores with their 95% CI and β-coefficient (95% CI) per walking pace category (slow/average/brisk) and Delta (95% CI); *p*-value between walking pace categories was estimated by linear regression analysis. Slow pace was used as the reference group [18]. 

All analyses were incrementally adjusted according to different confounding factors, and included four models: Model 0: unadjusted; Model 1: adjusted by sociodemographic factors: age, sex (female/male), education level (≤ 8 years/9–12 years/>12 years) and place of residence (urban/rural); Model 2: adjusted by model 1 plus BMI; and Model 3: adjusted by Model 2 plus lifestyle factors (smoking, alcohol use, fruit and vegetable intake, hours of sleep, physical activity, and sedentary time). Statistically significant values were considered when *p* < 0.05. All statistical analyses were performed with STATA 15 software (Statacorp; College Station, TX, USA).

## 3. Results

### 3.1. Characteristic of the Sample According to Walking Pace

A total of 17.9% of participants self-reported a slow walking pace, whereas 27.4% reported a brisk walking pace. Compared with brisk walkers, those who reported being slow walkers were older (39.1 and 55.7 years old, respectively). Participants in the slow walking pace category were women (60.8 2%) and had a lower education level (36.8% had ≤8 years). Slow walkers reported consuming more salt (9.5 g/day) compared with average and brisk walkers. The rates of total PA, transport-related PA, moderate intensity of PA, and vigorous intensity of PA were lower in slow walkers. Additionally, slow walkers had a higher prevalence of physical inactivity, sedentary time, central obesity, and obesity status compared with brisk walkers (Table 1).

### 3.2. Associations between Walking Pace and Cardiometabolic Health Markers

Associations between walking pace categories and cardiometabolic health markers are shown in Table 2. Compared with slow walkers, people who had an average or a brisk walking pace had lower levels of glycaemia (95.21 and 91.90 mg/dL, respectively), HbA1c (6.13 and 5.76, respectively), GGT (29.89 and 28.47 U/L, respectively), systolic blood pressure (122.53 and 120.26 mm/Hg, respectively), and diastolic blood pressure (74.17 and 73.34 mm/Hg, respectively) (Model 0, Table 2). Moreover, average and brisk walkers had higher Vitamin D3 levels (20.16 and 20.40 ng/mL, respectively) and Vitamin D2 + D3 levels (20.22 and 20.45 ng/mL, respectively) compared with slow walkers. However, after adjusting the model to include the confounding factors related to sociodemographic background, nutritional status, and lifestyle, the associations decreased but remained for glycaemia, HbA1c, and systolic blood pressure (Model 3, Table 2).

On the other hand, lower trend values for glycaemia (p-trend = −1.53 (−2.85; −0.22); *p* = 0.022), HbA1c (p-trend = −0.19 (−0.30; −0.68); *p* = 0.002) and systolic blood pressure (p-trend = −1.45 (−2.56; −0.33); *p* = 0.011) were observed for each increase in walking pace category. In addition, brisk walkers had lower levels of glycaemia (*p* = 0.049) and HbA1c (*p* = 0.002), and both average and brisk walkers had lower systolic blood pressure (0.006) compared with slow walkers (Model 3, Figure 1).

### 3.3. Associations between Walking Pace Categories and Metabolic Lipid Profile Outcomes

For the unadjusted model, average and brisk walkers had lower levels of VLDL-C (28.20 and 26.16 mg/dL, respectively) compared with slow walkers. However, after adjusting Model 3, these differences were not maintained (Table 3) and changes in trend values for each increase in walking pace category were not observed (Model 3, Figure 2).

## 4. Discussion

The aim of this study was to analyze the association between different walking pace categories and cardiometabolic health markers in the adult Chilean population. The main results of this study indicated that people who had a brisk walking pace possessed lower levels of glycaemia, HbA1c, GGT, systolic and diastolic blood pressure, and higher levels of vitamin D3 compared with those who had a slow walking pace. In addition, walking pace was associated with lower levels of VLDL-C for participants with a slow walking pace. However, after adjusting the model for confounding factors—sociodemographic background, nutritional status, and lifestyle—the levels of glycaemia, HbA1c and systolic blood pressure were maintained.

The associations identified between walking pace categories and blood glucose levels are consistent with existing evidence on this topic. Considering previous research, people who walk at a fast average pace have lower fasting glucose and HbA1c levels compared with slow walkers (glucose: average: 95.8; fast: 94.1; slow: 97.2) (HbA1c: average: 6.19%; fast: 5.88%; slow: 6.24%) [13]. In addition, it is established that a ≥1% decrease in the HbA1c value (7.3 ± 1.2; 7.1 ± 1.0) is a determining factor in changes in the walking pace because poor glycemic control is significantly associated with an increase in walking pace [26]. In the same way, people with high glycemic levels have a significantly slower walking pace (0.96 ± 0.02 m/s) than those without high glycemic levels (1.08 ± 0.01 m/s) [20]. Regarding the behavior of lipid profiles and gait walking pace, the existing evidence presents different results to those of this research. A study showed that individuals who reported walking at a slow pace had a higher concentration of triglycerides (Δ −2.32 mg/dl [95% IC: −4.24; −0.34], *p* = 0.022) and a lower concentration of HDL cholesterol compared with those who reported walking at a fast pace [27]. Other publications on this topic show similar results to the latter, where the basal lipid profile is more favorable in people who have a faster walking pace than those who walk slower (HDL-C 1.45 ± 0.39 vs. 10.24 ± 0.36 mmol/L, *p* = 0.14; LDL-C 3.4 ± 0.5 vs. 3.7 ± 0.8 mmol/L, *p* = 0.22; and triglycerides 1.1 ± 0.5 vs. 1.5 ± 0.9 mmol/L, *p* = 0.12, respectively) [28].

On the other hand, blood pressure was an indicator that presented changes when participants performed a fast walking pace. This is compared with a study of (*n* = 191) participants with a follow-up of 30 years, which indicated that high systolic and diastolic BP is associated with a slower walking pace. It was concluded that exposure to higher BP levels from youth to middle age is associated with a slower walking pace in midlife [29]. Moreover, an association was found between slow walking paces and low vitamin D levels and high levels of IL-6 [28]. This raises the concern that inflammation may be a factor to consider because it can affect how vitamin D works in the body [30]. Although most previous studies have focused on cardiovascular diseases only, the combination of walking pace and grip strength suggests a stronger association with health outcomes than in isolation [31].

### 4.1. How Does This Research Contribute to Society and Science?

This study reinforces the idea that a higher walking pace is associated with better cardiometabolic health and a better lipid profile [13,32]. In addition, this study allows the identification of walking pace as a physical-health marker associated with cardiometabolic health markers and lipid profile. In this context, the early assessment of walking pace could serve as a marker of cardiovascular and metabolic risk in adults [33]. It is highlighted that this study contributes to knowledge production on walking pace, which has not received enough scientific attention despite its associations with cardiometabolic health [34]. The scientific community is welcome to continue studying this topic thoroughly to improve our understanding regarding the implications of the association of walking pace with cardiovascular health and metabolic markers, as well as our understanding of how walking pace can help in the assessment and early screening of disorders, and of cardiovascular and lipid profiles.

In addition, walking pace has been studied as a factor associated with severity and mortality in people affected by COVID-19. Indeed, it has been observed that people who had a slow walking pace presented a significantly higher risk of severe symptoms [35] and a higher probability of severe infection and mortality due to COVID-19 [36] compared with those who had an average or brisk walking pace.

On the other hand, in practical terms, a walking pace assessment could be used by health professionals as a tool to identify patients who have a higher risk of contracting chronic non-communicable diseases [37]. Therefore, walking pace has been suggested as part of a screening questionnaire in primary care settings to identify high-risk patients. Moreover, walking pace could serve as an easy-to-apply and low-cost screening tool, with important prediction abilities for cardiovascular diseases, cancer, and premature mortality [38]. Walking pace could be a simple, safe, free, and feasible way to increase PA and improve physical fitness in patients with cardiometabolic diseases.

### 4.2. Limitations and Future Research

The results of this study can be generalized to the Chilean population with more confidence than previous estimates due to the representativeness of the population sample. More importantly, this is the first study to report associations between walking pace and cardiometabolic health. However, it is not exempt from limitations. Firstly, self-reported data on walking pace were used due to a lack of objectively measured summary-level data, which might lead to a misclassification bias. However, self-reported walking pace is highly correlated with actual measured walking pace [39]. Walking pace was determined through the question “How would you describe your usual walking pace?” This question, to the best of our knowledge, is not validated, even though it is widely used to measure self-reported walking pace [12,13,14,19,20,40]. The effects of confounding factors regarding waist circumference and physical activity could not be removed, as this might have resulted in multicollinearity issues. Other potential confounders such as the presence of comorbidities, the intensity of PA, muscle strength, and balance, were not collected in the CNHS. In addition, the possible effects of multimorbidity on our findings could not be eliminated: people who reported a slow walking pace may also have other chronic diseases that may limit their walking pace. Finally, due to the design of our study, our results cannot prove causality. However, previous evidence from randomized controlled trials suggests that walking is associated with a better glycemic control in healthy and diabetic patients [41].

## 5. Conclusions

Regardless of the different sociodemographic and lifestyle characteristics of this adult cohort, the Chilean National Health Survey revealed that the adult population who had a ‘brisk’ walking pace possessed lower levels of glycaemia, HbA1c and systolic blood pressure, showing a healthier profile overall. Further investigation is needed to thoroughly understand the associations between other physical-capability markers and health outcomes in Chile.

## Figures and Tables

**Figure 1 ijerph-20-05490-f001:**
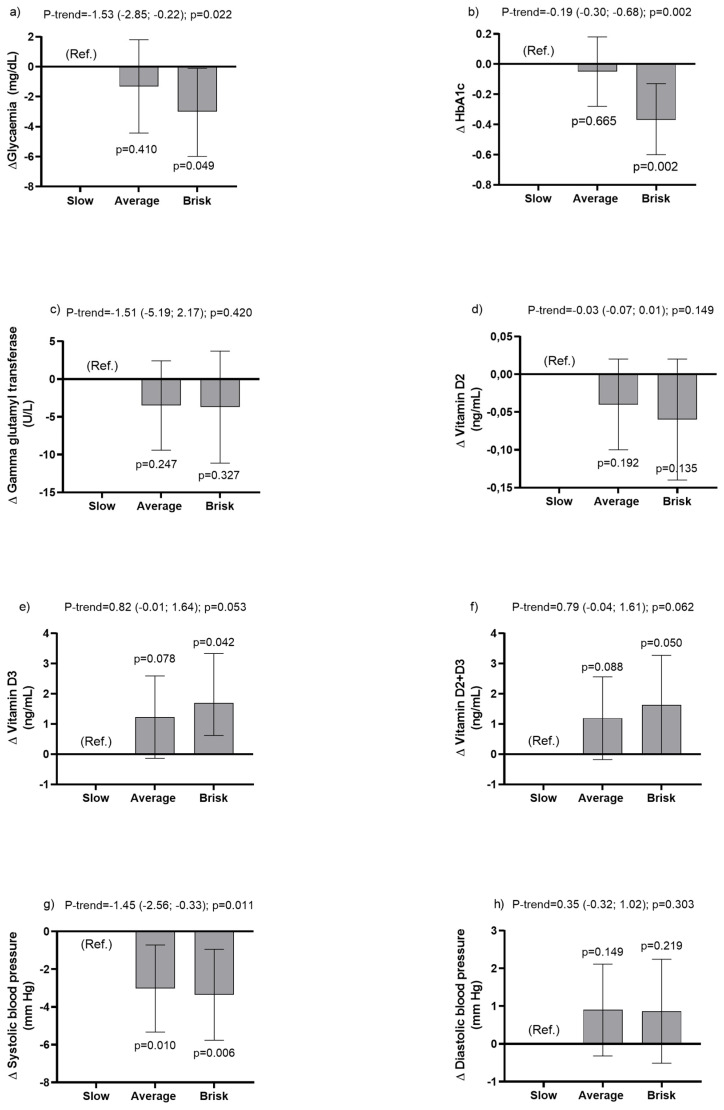
Association of walking pace with cardiometabolic health markers: glycaemia (panel (**a**)), HbA1c (panel (**b**)), gamma glutamyl transferase (panel (**c**)), vitamin D2 (panel (**d**)), vitamin D3 (panel (**e**)), vitamin D3 + D2 (panel (**f**)), systolic BP (panel (**g**)), and diastolic BP (panel (**h**)). Data presented as β-coefficient and their 95% confidence intervals (CI) by walking pace category estimated by linear regression analysis. Slow pace was used as the reference group (Ref.). Graphs shown were obtained from adjusted Model 3.

**Figure 2 ijerph-20-05490-f002:**
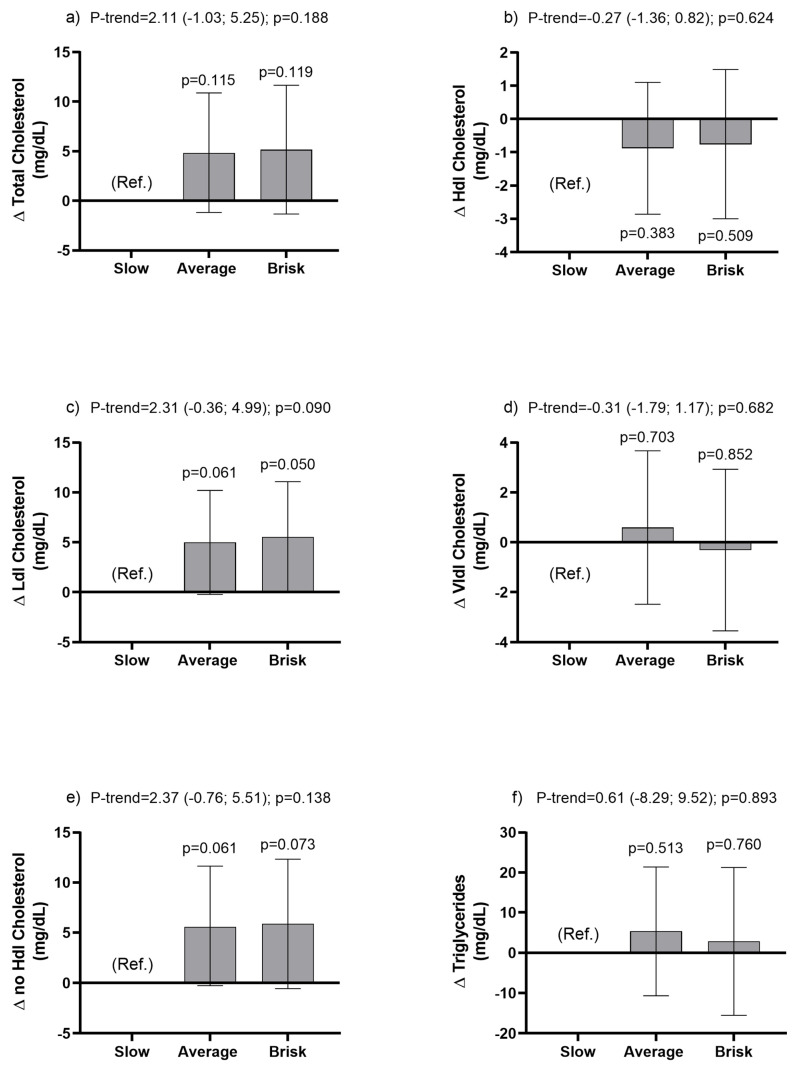
Association of walking pace with lipid profile. (panel (**a**)), Total cholesterol (panel (**b**)), Hdl cholesterol (panel (**c**)), Ldl cholesterol (panel (**d**)), Vldl cholesterol (panel (**e**)), no Hdl cholesterol (panel (**f**)), Triglycerides). Data presented as β-coefficient and their 95% CI estimated by linear regression analysis. Slow pace was used as the reference group (Ref.). Graphs shown were obtained from adjusted Model 3.

**Table 1 ijerph-20-05490-t001:** Characteristic of the sample according to walking pace.

Variables	*Walking Pace*		
	*Slow Pace*	*Average Pace*	*Brisk Pace*
***n* (%)**	17.9 (16.3; 19.6)	54.7 (52.3; 57.0)	27.4 (25.3; 29.7)
**Estimation sample ****	2.585.018	7.903.904	3.965.239
**Sociodemographic**			
Age (years) *	55.7 (53.6; 57.8)	40.1 (40.0; 42.0)	39.1 (37.7; 40.5)
Sex (%)			
Women	60.8 (55.7; 65.7)	47.4 (44.2; 50.5)	51.1 (46.3; 55.9)
Men	39.2 (34.3.3; 44.3)	52.6 (49.5; 55.8)	48.1 (44.1; 53.7)
Place of residence (%)			
Urban	84.7 (81.6; 87.3)	88.7 (87.1; 90.2)	92.5 (90.5; 94.1)
Rural	15.3 (12.6; 18.3)	11.2 (9.8; 12.8)	7.4 (5.8; 9.5)
Education level (%)			
≤8 years	36.8 (32.4; 41.4)	13.0 (11.1; 15.0)	9.6 (7.3; 12.6)
9–12 years	48.0 (43.0; 53.1)	59.1 (55.8; 62.2)	55.1 (50.3; 59.9)
>12 years	15.2 (11.6; 19.6)	28.0 (25.0; 31.1)	35.2 (30.7; 40.1)
**Lifestyle**			
Smoking (%)			
Regular smoker	18.7 (14.9; 23.2)	26.3 (23.5; 29.4)	24.1 (20.2; 28.5)
Occasional smoker	6.4 (4.1; 9.7)	9.3 (7.5; 11.5)	7.8 (5.6; 10.7)
Ex-smoker	27.3 (23.4; 31.6)	24.7 (21.9; 27.6)	26.5 (22.4; 31.0)
Non-smoking	47.6 (42.5; 52.6)	39.7 (36.7; 42.7)	41.6 (37.0; 46.3)
Alcohol use (%)			
High consumption (test AUDIT)	4.3 (1.8; 10.2)	5.0 (4.0; 7.3)	6.1 (3.7; 9.8)
F and V intake (%)			
Eats less than 5 F and V	87.7 (84.2; 90.5)	86.2 (83.6; 88.5)	81.5 (77.4; 85.0)
Salt intake (g/day) *	9.5 (9.2; 9.8)	9.1 (8.9; 9.2)	8.9 (8.6; 9.2)
**Hours of sleep (%)**			
≤6 h	28.9 (24.4; 33.8)	20.2 (17.7; 23.0)	24.1 (20.2; 28.4)
7–8 h	45.2 (40.4; 50.3)	54.2 (50.6; 57.3)	52.5 (47.8; 57.3)
≥9 h	25.9 (21.6; 30.6)	25.6 (23.0; 28.5)	23.4 (19.7; 27.6)
**Adiposity**			
Body weight (kg) *	76.3 (74.6; 78.1)	75.7 (74.7; 76.7)	74.7 (73.3; 76.2)
BMI (kg/m^2^) *	30.2 (29.6; 30.9)	28.4 (28.1; 28.7)	27.7 (27.3; 28.1)
Nutritional status (%)			
Underweight	1.1 (0.0; 2.6)	1.5 (0.1; 2.6)	0.1 (0.0; 0.2)
Normal	21.8 (17.8; 26.4)	24.2 (21.6; 27.0)	27.0 (23.0; 31.4)
Overweight	29.9 (25.6; 34.6)	41.7 (38.6; 44.9)	42.4 (37.7; 47.3)
Obese	47.2 (42.1; 52.3)	32.6 (29.7; 35.6)	29.8 (25.6; 34.3)
Waist circumference (cm) *	98.3 (96.9; 99.8)	92.6 (91.7; 93.5)	91.3 (90.1; 92.4)
Central obesity (cm) (%)			
>102 cm men, >88 cm women	59.0 (53.9; 63.9)	41.3 (38.3; 44.4)	39.2 (34.6; 43.9)
**Physical activity ***			
Total PA (MET/min/day)	771.1 (633; 909)	1200 (1100; 1301)	1412 (1232; 1591)
Transport PA (min/day)	46.6 (32.9; 60.4)	72.5 (64.2; 80.7)	78.5 (65.8; 91.1)
Moderate PA (min/day)	63.5 (49.9; 77.1)	93.3 (82.2; 104.9)	122.8 (102.8; 1428)
Vigorous PA (min/day)	41.3 (27.0; 55.6)	67.2 (57.9; 76.8)	75.8 (60.3; 91.4)
Sedentary time (min/day)	214 (196; 231)	199 (188; 210)	208 (190; 226)
Physical inactivity (%)	38.8 (34.1; 43.6)	23.9 (21.3; 26.7)	18.9 (15.5; 22.8)

Caption: Data presented by average and 95% CI for continuous variables (*) and in % and 95% CI for categorical variables. ** A logarithm of extension and amplification of the sample was used. F and V = Fruit and Vegetable; AUDIT = Alcohol Use Disorders Identification Test; BMI = Body Mass Index; PA = Physical Activity.

**Table 2 ijerph-20-05490-t002:** Associations between walking pace and cardiometabolic health markers.

Variables	Slow Pace Mean (95% CI)	Average Pace Mean (95% CI)	Brisk Pace Mean (95% CI)	Delta (95% CI); *p*-Value
**Glycaemia (mg/dL)**				
Model 0	102.95 (99.86; 106.04)	95.21 (93.93; 96.49)	91.90 (90.73; 93.06)	−5.24 (−6.73; −3.74); *p* < 0.0001
Model 1	98.38 (95.54; 101.21)	95.82 (94.58; 97.07)	93.53 (92.36; 94.70)	−2.40 (−7.77; −1.04); *p* = 0.001
Model 2	97.61 (94.78; 100.44)	95.85 (94.62; 97.08)	93.96 (92.80; 95.12)	−1.83 (−3.21; −0.46); *p* = 0.009
Model 3	97.8 (94.42; 99.95)	95.87 (94.63; 97.12)	94.18 (93.00; 95.35)	−1.53 (−2.85; −0.22); *p* = 0.022
**Glycosylated haemoglobin (HbA1c)**				
Model 0	6.47 (6.28; 6.66)	6.13 (6.00; 6.27)	5.76 (5.64; 5.87)	−0.36 (−0.47; −0.25); *p* < 0.0001
Model 1	6.27 (6.09; 6.44)	6.19 (6.06; 6.31)	5.87 (5.74; 6.01)	−0.20 (−0.31; −0.08); *p* = 0.001
Model 2	6.26 (6.08; 6.45)	6.19 (6.06; 6.31)	5.88 (5.74; 6.01)	−0.20 (−0.31; −0.08); *p* = 0.001
Model 3	6.25 (6.06; 6.43)	6.20 (6.07; 6.32)	5.88 (5.75; 6.01)	−0.19 (−.030; −0.68); *p* = 0.002
**Gamma glutamyl transferase (GGT) (U/L)**				
Model 0	38.58 (33.45; 43.72)	29.89 (27.42; 32.36)	28.47 (23.96; 32.98)	−4.50 (−7.91; −1.10); *p* = 0.009
Model 1	34.27 (29.14; 39.40)	30.20 (27.70; 32.69)	29.78 (25.04; 34.52)	−1.89 (−5.64; 1.86); *p* = 0.325
Model 2	33.42 (28.15; 38.68)	30.24 (27.78; 32.71)	30.35 (25.59; 35.11)	−1.20 (−5.04; 2.63); *p* = 0.539
Model 3	33.78 (28.64; 38.92)	30.29 (27.83; 32.75)	30.07 (25.46; 34.68)	−1.51 (−5.19; 2.17); *p* = 0.420
**Vitamin D2 (ng/mL)**				
Model 0	0.10 (0.04; 0.15)	0.06 (0.01; 0.10)	0.04 (0.01; 0.07)	−0.03 (−0.06; 0.01); *p* = 0.103
Model 1	0.08 (0.03; 0.14)	0.06 (0.02; 0.11)	0.04 (0.01; 0.08)	−0.02 (−0.05; 0.01); *p* = 0.229
Model 2	0.09 (0.03; 0.15)	0.06 (0.02; 0.10)	0.04 (0.00; 0.08)	−0.03 (−0.06; 0.01); *p* = 0.166
Model 3	0.10 (0.03; 0.16)	0.06 (0.02; 0.10)	0.04 (0.00; 0.07)	−0.03 (−0.07; 0.01); *p* = 0.149
**Vitamin D3 (ng/mL)**				
Model 0	18.40 (17.43; 19.38)	20.16 (19.42; 20.90)	20.40 (19.11; 21.70)	0.99 (0.17; 1.81); *p* = 0.018
Model 1	18.36 (17.28; 19.44)	20.4 (19.31; 20.77)	20.62 (19.37; 21.88)	1.09 (0.25; 1.93); *p* = 0.011
Model 2	18.44 (17.34; 19.53)	20.03 (19.30; 20.76)	20.60 (19.36; 21.85)	1.05 (0.21; 1.88); *p* = 0.015
Model 3	18.74 (17.59; 19.90)	19.97 (19.25; 20.69)	20.44 (19.23; 21.64)	0.82 (−0.01; 1.64); *p* = 0.053
**Vitamin D2 + D3 (ng/mL)**				
Model 0	18.50 (17.52; 19.48)	20.22 (19.48; 20.96)	20.45 (19.15; 21.75)	0.96 (0.14; 1.78); *p* = 0.021
Model 1	18.45 (17.37; 19.53)	20.10 (19.37; 20.83)	20.67 (19.41; 21.92)	1.07 (0.23; 1.91); *p* = 0.012
Model 2	18.53 (17.43; 19.62)	20.09 (19.36; 20.82)	20.64 (19.39; 21.89)	1.02 (0.18; 1.86); *p* = 0.017
Model 3	18.84 (17.69; 20.00)	20.03 (19.31; 20.75)	20.48 (19.27; 21.68)	0.79 (−0.04; 1.61); *p* = 0.062
**Systolic blood pressure (mm/Hg)**				
Model 0	133.03 (130.67; 135.40)	122.53 (121.38; 123.68)	120.26 (118.92; 121.60)	−5.90 (−7.19; −4.62); *p* < 0.0001
Model 1	126.96 (124.91; 129.01)	123.30 (122.33; 124.27)	122.56 (121.35; 123.77)	−1.97 (−3.12; −0.83); *p* = 0.001
Model 2	126.22 (124.22; 128.21)	123.29 (122.34; 124.23)	122.90 (121.72; 124.08)	−1.45 (−2.56; −0.35); *p* = 0.010
Model 3	126.28 (124.26; 128.30)	123.25 (122.32; 124.19)	122.92 (121.74; 124.12)	−1.45 (−2.56; −0.33); *p* = 0.011
**Diastolic blood pressure (mm/Hg)**				
Model 0	75.44 (74.35; 76.53)	74.17 (73.51; 74.84)	73.34 (72.42; 74.25)	−1.03 (−1.73; −0.32); *p* = 0.004
Model 1	74.01 (72.82; 75.17)	74.35 (73.75; 74.95)	73.94 (73.06; 74.81)	−0.10 (−0.81; 0.62); *p* = 0.792
Model 2	73.26 (72.21; 74.32)	74.37 (73.81; 74.93)	74.31 (73.48; 75.14)	0.43 (−0.24; 1.10); *p* = 0.206
Model 3	73.43 (72.36; 74.50)	74.33 (73.77; 74.88)	74.29 (73.47; 75–11)	0.35 (−0.32; 1.02); *p* = 0.303

Caption: Data presented as adjusted means and their 95% confidence intervals (CI) and estimated by linear regression analysis. *p* value indicates significant differences between groups. Statistical analyses were incrementally adjusted and included four models: Model 0 was unadjusted; Model 1 was adjusted by sociodemographic factors and place of residence; Model 2 was adjusted by Model 1 and additionally adjusted by BMI; Model 3 was adjusted by Model 1 and 2 and additionally adjusted by lifestyle factors.

**Table 3 ijerph-20-05490-t003:** Associations between walking pace categories and metabolic lipid profile outcomes.

Variables	Slow Pace Mean (95% IC)	Average Pace Mean (95% IC)	Brisk Pace Mean (95% IC)	Delta (95% CI); *p*-Value
**Total Cholesterol (mg/dL)**				
Model 0	179.94 (175.00; 184.89)	176.66 (173.66; 179.66)	175.85 (171.98; 179.73)	−1.86 (−4.95; 1.23); *p* = 0.239
Model 1	173.20 (167.97; 178.43)	177.69 (174.79; 180.58)	177.63 (173.79; 181.47)	1.77 (−1.39; 4.93); *p* = 0.272
Model 2	172.30 (167.01; 177.60)	177.73 (174.86; 180.59)	178.15 (174.32; 181.97)	2.42 (−0.76; 5.60); *p* = 0.136
Model 3	172.83 (167.64; 178.02)	177.68 (174.81; 180.54)	177.99 (174.21; 181.77)	2.11 (−1.03; 5.25); *p* = 0.188
**HDL Cholesterol (mg/dL)**				
Model 0	47.30 (45.55; 49.04)	46.27 (45.27; 47.27)	47.52 (45.98; 49.06)	0.28 (−0.88; 1.45); *p* = 0.633
Model 1	46.34 (44.43; 48.25)	46.59 (45.64; 47.53)	47.44 (46.05; 48.83)	0.61 (−0.54; 1.76); *p* = 0.297
Model 2	47.35 (45.55; 49.15)	46.55 (45.65; 47.45)	46.92 (45.60; 48.24)	−0.09 (−1.19; 1.00); *p* = 0.866
Model 3	47.49 (45.70; 49.28)	46.61 (45.71; 47.50)	46.73 (45.44; 48.02)	−0.27 (−1.36; 0.82); *p* = 0.624
**LDL Cholesterol (mg/dL)**				
Model 0	102.64 (98.46; 106.81)	101.74 (98.95; 104.53)	101.21 (98.10; 104.32)	−0.69 (−3.24; 1.87); *p* = 0.599
Model 1	98.18 (93.77; 102.59)	102.43 (99.76; 105.10)	102.46 (99.36; 105.55)	1.73 (−0.87; 4.33); *p* = 0.193
Model 2	97.20 (92.75; 101.65)	102.46 (99.81; 105.11)	102.97 (99.87; 106.06)	2.42 (−0.20; 5.03); *p* = 0.070
Model 3	97.42 (92.95; 101.89)	102.42 (99.76; 105.07)	102.97 (99.84; 106.09)	2.31 (−0.36; 4.99); *p* = 0.090
**VLDL Cholesterol (mg/dL)**				
Model 0	29.67 (27.04; 32.31)	28.20 (26.77; 29.63)	26.16 (24.64; 27.68)	−1.80 (−3.24;−0.36); *p* = 0.014
Model 1	28.46 (25.53; 31.40)	28.23 (26.87; 29.58)	26.70 (25.20; 28.19)	−1.01 (−2.56; 0.55); *p* = 0.205
Model 2	27.63 (24.79; 30.46)	28.26 (26.93; 29.58)	27.20 (25.74; 28.66)	−0.38 (−1.88; 1.11); *p* = 0.617
Model 3	27.62 (24.82; 30.41)	28.21 (26.92; 29.52)	27.31 (25.84; 28.77)	−0.31 (−1.79; 1.17); *p* = 0.682
**No HDL Cholesterol (mg/dL)**				
Model 0	132.65 (127.73; 137.58)	130.35 (127.20; 133.51)	128.32 (124.37; 132.27)	−2.15 (−5.28; 0.98); *p* = 0.178
Model 1	126.88 (121.59; 132.16)	131.05 (128.06; 134.05)	130.18 (126.30; 134.06)	1.16 (−2.02; 4.34); *p* = 0.475
Model 2	124.96 (119.76; 130.16)	131.13 (128.22; 134.04)	131.21 (127.42; 135.00)	2.51 (−0.63; 5.65); *p* = 0.117
Model 3	125.34 (120.24; 130.45)	131.04 (128.14; 133.93)	131.24 (127.45; 135.02)	2.37 (−0.76; 5.50); *p* = 0.138
**Triglycerides (mg/dL)**				
Model 0	149.63 (136.10; 163.17)	144.37 (135.59; 152.15)	135.83 (125.27; 146.39)	−7.15 (−15.60; 1.30); *p* = 0.097
Model 1	143.41 (128.23; 158.58)	144.47 (137.06; 151.88)	138.71 (127.91; 149.51)	−3.01 (−12.35; 6.32); *p* = 0.527
Model 2	138.54 (123.91; 153.17)	144.68 (137.45; 151.92)	141.48 (130.90; 152.06)	0.53 (−8.64; 9.69); *p* = 0.910
Model 3	139.03 (124.59; 153.46)	144.48 (137.32; 151.45)	141.89 (131.48; 152.31)	0.61 (−8.29; 9.52); *p* = 0.893

Caption: Data presented as adjusted means and their 95% confidence intervals (CI) and estimated by linear regression analysis. *p* value indicates significant differences between groups. Statistical analyses were incrementally adjusted and included four models: Model 0 was unadjusted; Model 1 was adjusted by sociodemographic factors and place of residence; Model 2 was adjusted by Model 1 and additionally adjusted by BMI; Model 3 was adjusted by Model 1 and 2 and additionally adjusted by lifestyle factors.

## Data Availability

Data are available upon request due to ethical and privacy restrictions.

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
