# Peer review of "Brisk Walking Pace Is Associated with Better Cardiometabolic Health in Adults: Findings from the Chilean National Health Survey 2016–2017"

_ijerph, 2023, doi:10.3390/ijerph20085490_

Round 1

Reviewer 1 Report

I will use your words to write my comments. I agree that the national representativeness of the population is one of the main straight of this study. However, is not exempt from limitations. Firstly, walking pace was self-reported  and other potential confounders such as intensity of PA, muscle strength, and balance, were not collected in the CNHS. In addition, you cannot eliminate the possible effect of multimorbidity on our findings: people who reported a slow walking pace may also have other chronic diseases that may limit their walking pace. Finally, due to design of study, your results cannot prove causality. More investigation is needed to understand in-deep the associations between other physical capability markers and different health outcomes in Chile.

Author Response

REVIEWER 1

Comment 1: I will use your words to write my comments. I agree that the national representativeness of the population is one of the main straight of this study. However, is not exempt from limitations. Firstly, walking pace was self-reported and other potential confounders such as intensity of PA, muscle strength, and balance, were not collected in the CNHS. In addition, you cannot eliminate the possible effect of multimorbidity on our findings: people who reported a slow walking pace may also have other chronic diseases that may limit their walking pace. Finally, due to design of study, your results cannot prove causality. More investigation is needed to understand in-deep the associations between other physical capability markers and different health outcomes in Chile.

Answer 1: Thank you very much for the comment regarding the limitations. We are aware of all the limitations of the study. These limitations are typical in this type of study, and it can be read that our limitations are not different from similar studies that have focused on studying walking pace measured through self-report and that have analyzed its association with cardiovascular diseases, diabetes, stroke or all the causes of death. Examples of articles that have similar limitations are shared below.

  • Walking Pace Is Associated with Lower Risk of All-Cause and Cause-Specific Mortality

https://pubmed.ncbi.nlm.nih.gov/30303933/#:~:text=Purpose%3A%20Walking%20pace%20is%20associated,time%20walked%20are%20currently%20unknown.

  • Association of Self-reported Walking Pace With Type 2 Diabetes Incidence in the UK Biobank Prospective Cohort Study

https://pubmed.ncbi.nlm.nih.gov/36058577/

  • Association Between Walking Pace and Stroke Incidence: Findings From the UK Biobank Prospective Cohort Study

https://pubmed.ncbi.nlm.nih.gov/32299326/

  • Self-rated walking pace and all-cause, cardiovascular disease and cancer mortality: individual participant pooled analysis of 50 225 walkers from 11 population British cohorts

https://pubmed.ncbi.nlm.nih.gov/29858463/

  • Self-Reported Walking Pace and Risk of Cardiovascular Diseases: A Two-Sample Mendelian Randomization Study

https://pubmed.ncbi.nlm.nih.gov/35783285/

Even authors of this manuscript have published similar articles in this journal:

  • Association between Walking Pace and Diabetes: Findings from the Chilean National Health Survey 2016–2017

https://pubmed.ncbi.nlm.nih.gov/32722215/

However, the limitations section of this article has been strengthened.

Reviewer 2 Report

In 4.21 it is acknowledged the self-reporting of walking pace is a limitation and a study is referenced in regard to this question being a significant predictor of health. However, there is no reference to the validity and reliability of this self-reported measure. 

The brisk walkers had higher total, moderate, and vigorous PA and therefore a higher volume of activity. How much did this contribute compared to the intensity of the walking?

Author Response

REVIEWER 2

Comment 1: In 4.21 it is acknowledged the self-reporting of walking pace is a limitation and a study is referenced in regard to this question being a significant predictor of health. However, there is no reference to the validity and reliability of this self-reported measure. 

The brisk walkers had higher total, moderate, and vigorous PA and therefore a higher volume of activity. How much did this contribute compared to the intensity of the walking?

Answer 1: Thank you very much for the comment regarding the limitations. As indicated in the methodology, this study used data from the 2016-2017 National Health Survey of Chile. Walking pace was determined through the following question “How would you describe your usual walking pace?” from which participants were asked to select one of the following walking pace categories (slow, average, or brisk pace). Although this question has been used in multiple studies to measure self-reported walking pace, we do not know if it is validated.

We agree that a non-objective measure was used to assess walking pace. In this study, walking pace was measured through self-report. This strategy is widely used in the literature when large samples are analyzed, as it is the case of this study. We are aware that self-reported walking pace measure is a limitation of this study, but it can be read that our limitation is not different from similar studies that have focused on studying walking pace measured through self-report and that have analyzed its association with cardiovascular diseases, diabetes, stroke or all the causes of death. Examples of articles that have similar limitations are shared below.

  • Walking Pace Is Associated with Lower Risk of All-Cause and Cause-Specific Mortality

https://pubmed.ncbi.nlm.nih.gov/30303933/#:~:text=Purpose%3A%20Walking%20pace%20is%20associated,time%20walked%20are%20currently%20unknown.

  • Association of Self-reported Walking Pace With Type 2 Diabetes Incidence in the UK Biobank Prospective Cohort Study

https://pubmed.ncbi.nlm.nih.gov/36058577/

  • Association Between Walking Pace and Stroke Incidence: Findings From the UK Biobank Prospective Cohort Study

https://pubmed.ncbi.nlm.nih.gov/32299326/

  • Self-rated walking pace and all-cause, cardiovascular disease and cancer mortality: individual participant pooled analysis of 50 225 walkers from 11 population British cohorts

https://pubmed.ncbi.nlm.nih.gov/29858463/

  • Self-Reported Walking Pace and Risk of Cardiovascular Diseases: A Two-Sample Mendelian Randomization Study

https://pubmed.ncbi.nlm.nih.gov/35783285/

Even authors of this manuscript have published similar articles in this journal:

  • Association between Walking Pace and Diabetes: Findings from the Chilean National Health Survey 2016–2017

https://pubmed.ncbi.nlm.nih.gov/32722215/

Finally, it should be mentioned that the regression model used is adjusted for confounding variables, including lifestyle factors such as smoking, alcohol use, fruit and vegetable intake, sleep hours, physical activity and sedentary time, so the results obtained are independent of these variables.

However, the limitations section of this article has been strengthened.

Reviewer 3 Report

This original article evaluates the possible correlation between difference walking pace categories and cardiometabolic health biomarkers. The strong point of the study is the sample size but it presents a series of inaccuracies in the methodology of the study design.  The selected age range is too wide (15-90 years ). In fact, there are too many age-related variables that impact on the cardiometabolic profile. Therefore, the age range should be reviewed, selecting the range 18-70 years. In the light of this new age range, the statistics should be reviewed.Furthermore, the authors use a subject and non-objective parameter to evaluate walking pace. Furthermore, confounding factors such as the presence of comorbidities were not taken into account in the statistical model. 

Minor comments.

There are several grammar errors in the text, so proofreading of the English is required.

in the introduction there are a series of inaccuracies on non-communicable chronic degenerative diseases, on risk factors, etc. Rewrite the introduction in a correct and more focused way.

Author Response

Reviewer 3

Comment 1: This original article evaluates the possible correlation between difference walking pace categories and cardiometabolic health biomarkers. The strong point of the study is the sample size but it presents a series of inaccuracies in the methodology of the study design.  

The selected age range is too wide (15-90 years ). In fact, there are too many age-related variables that impact on the cardiometabolic profile. Therefore, the age range should be reviewed, selecting the range 18-70 years. In the light of this new age range, the statistics should be reviewed.

Answer 1: Thank you very much for the comment. However, we disagree on this point. We believe that the analysis of a wider age range is a strength and not a limitation.

Most of the studies that analyze walking pace as a physical health marker use narrower age ranges, so it is not possible to know if this association occurs in a broader age range.

  • 40-69 years; doi: 1249/MSS.0000000000001795
  • 37-73 years; doi: 10.1161/STROKEAHA.119.028064.Epub 2020 Apr 17

Besides that, a study published in this same journal used wider age ranges to analyze gait speed as a physical health marker.

  • 15-90 years: doi:10.3390/ijerph17155341

Finally, it should be mentioned that the regression model used is adjusted for confounding variables, including age, so the results obtained are independent of the age of the participants.

Comment 2: Furthermore, the authors use a subject and non-objective parameter to evaluate walking pace.

Answer 2: We agree that a non-objective measure was used to assess walking pace. In this study, walking pace was measured through self-report. This strategy is widely used in the literature when large samples are analyzed, as it is the case of this study. We are aware that self-reported walking pace is a limitation of this study, but it can be read that our limitation is not different from similar studies that have focused on studying walking pace measured through self-report and that have analyzed its association with cardiovascular diseases, diabetes, stroke or all the causes of death. Examples of articles that have similar limitations are shared below.

  • Walking Pace Is Associated with Lower Risk of All-Cause and Cause-Specific Mortality

https://pubmed.ncbi.nlm.nih.gov/30303933/#:~:text=Purpose%3A%20Walking%20pace%20is%20associated,time%20walked%20are%20currently%20unknown.

  • Association of Self-reported Walking Pace With Type 2 Diabetes Incidence in the UK Biobank Prospective Cohort Study

https://pubmed.ncbi.nlm.nih.gov/36058577/

  • Association Between Walking Pace and Stroke Incidence: Findings From the UK Biobank Prospective Cohort Study

https://pubmed.ncbi.nlm.nih.gov/32299326/

  • Self-rated walking pace and all-cause, cardiovascular disease and cancer mortality: individual participant pooled analysis of 50 225 walkers from 11 population British cohorts

https://pubmed.ncbi.nlm.nih.gov/29858463/

  • Self-Reported Walking Pace and Risk of Cardiovascular Diseases: A Two-Sample Mendelian Randomization Study

https://pubmed.ncbi.nlm.nih.gov/35783285/

Even authors of this manuscript have published similar articles in this journal:

  • Association between Walking Pace and Diabetes: Findings from the Chilean National Health Survey 2016–2017

https://pubmed.ncbi.nlm.nih.gov/32722215/

However, the limitations section of this article has been strengthened.

Comment 3: Furthermore, confounding factors such as the presence of comorbidities were not taken into account in the statistical model. 

Answer 3: We are aware that many confounding variables were not taken into account to fit the regression model. The reason for this omission is that these confounding factors were not measured in the survey that was analyzed (Chilean National Health Survey (CHNS) 2016-2017). This comment is considered in the limitations.

Minor comments.

Comment 4: There are several grammar errors in the text, so proofreading of the English is required.

in the introduction there are a series of inaccuracies on non-communicable chronic degenerative diseases, on risk factors, etc. Rewrite the introduction in a correct and more focused way.

Answer 4: The manuscript was fully reviewed by a university professor who is an expert in linguistics to eliminate any grammatical errors detected.